# Sleep Quality Should Be Assessed in Inpatient Rehabilitation Settings: A Preliminary Study

**DOI:** 10.3390/brainsci13050718

**Published:** 2023-04-25

**Authors:** Benjamin Winters, Dylan Serpas, Niko Fullmer, Katie Hughes, Jennifer Kincaid, Emily R. Rosario, Caroline Schnakers

**Affiliations:** 1Department of Psychology, University of California, Los Angeles, CA 90095, USA; 2Department of Psychology, University of South Florida, Tampa, FL 33620, USA; 3Research Institute, Casa Colina Hospital and Centers for Healthcare, Pomona, CA 91767, USA; 4Department of Nursing, Casa Colina Hospital and Centers for Healthcare, Pomona, CA 91767, USA; 5Respiratory Care Services, Casa Colina Hospital and Centers for Healthcare, Pomona, CA 91767, USA

**Keywords:** sleep quality, inpatient rehabilitation, sleep disorder, Pittsburgh sleep quality index

## Abstract

Objectives: In this preliminary, longitudinal study, our objective was to assess changes in sleep quality during an inpatient stay in a rehabilitation setting in the United States and to relate changes to patients’ demographic and clinical characteristics (i.e., age, gender, BMI, ethnicity, reason for hospitalization, pre-hospital living setting, prior diagnosis of sleep disorders, and mental health status). Methods: A total of 35 patients participated in this preliminary study (age = 61 ± 16 years old, 50% <65; BMI = 30 ± 7 kg/m^2^; 51% female; 51% Caucasian). The average length of hospitalization was 18 ± 8 days. Reasons for hospitalization included orthopedic-related issues (28%), spinal cord injury (28%), stroke (20%), and other (23%). In this sample, 23% had prior sleep disorders (mostly sleep apnea), and 60% came from an acute care unit. Patients’ sleep quality was assessed using the Pittsburgh sleep quality index (PSQI) at admission and before discharge. Demographic and medical data were collected. Patients’ mental health status was also assessed at the same intervals. Nighttime sound levels and the average number of sleep disturbances were also collected throughout the study (6 months). Results: Our data revealed that most patients had poor sleep (PSQI > 5) at admission (86%) and discharge (80%). Using a repeated ANOVA, a significant interaction was obtained between sleep quality and the presence of a diagnosed sleep disorder [F (1, 33) = 12.861, *p* = 0.001, η^2^*_p_* = 0.280]. The sleep quality of patients with sleep disorders improved over their stay, while the sleep of patients without such disorders did not. The mean nighttime sound collection level averages and peaks were 62.3 ± 5.1 dB and 86.1 ± 4.9 dB, respectively, and the average number of sleep disturbances was 2.6 ± 1.1. Conclusion: The improved sleep observed in patients with vs. without sleep disorders might be related to the care received for treating such disorders over the stay. Our findings call for the better detection and management of poor sleep in acute inpatient rehabilitation settings. Furthermore, if our findings are replicated in the future, studies on the implementation of quiet times for medical staff, patients, and family should be performed to improve sleep quality in the inpatient rehabilitation setting.

## 1. Introduction

Sleep is an essential component of health, especially after an initial illness or injury during rehabilitation. Studies have established how the deprivation of sleep affects body-wide metabolism and hormone regulation [1]. Furthermore, inadequate sleep has been shown to affect specific conditions and organ function, including hypertension [2], obesity [3], immune system response [4], cardiovascular disease [5], mood disorders [6], neurodegeneration [7], and dementia [8]. Sleep quality has also been related to demographics such as gender, age, or body mass index [9]. This understanding of the impact of sleep on health is particularly important to patients in a hospital setting. Over the past decade, a series of studies have investigated the interest in improving sleep quality in acute hospitals. Disrupted sleep is among the most common complaints of patients that survive a critical illness. Only 50% of sleep hours occur during the night with increased amounts of wakefulness and stage I sleep (drowsiness) and decreased amounts of SWS (slow wave sleep) and rapid eye movement (REM) sleep, which are critical in memory consolidation [10]. In the general population, sleep loss is a marker for mortality [11]. Reasons for sleep deprivation during recovery from acute illness and injury are multifactorial: type and severity of underlying illness, primary sleep disorders, pain, medical comorbidities, complications (such as respiratory dysfunction and congestive heart failure), medications (sedatives, analgesics), psychological factors (depression and anxiety) and also the environment itself (excessive noise and light as well as patient care activities) [12].

Scheduled rest and sleep seem to promote relaxation and reduction of stress levels [13]. However, even though interventions to promote inpatient rest and sleep have been successfully trialed in critical care units, scant research exists to indicate whether this understanding applies to the acute inpatient rehabilitation setting. Only a handful of studies indicated that sleep quality is particularly important for patients recovering from injury or illness in that setting. Sonmez et al. (2019) found that the presence of poor sleep quality is associated with reduced functional outcomes which further impairs the rehabilitation process and, accordingly, the health status in patients admitted for stroke [14]. Furthermore, poor self-reported sleep quality was related to depressive symptoms, pain, and predicted mortality within one year of inpatient post-acute rehabilitation among older adults [15,16,17]. Finally, Davis et al. (2021) found that patients in the acute rehabilitation unit experience sleep quality that matches their experience at home and exceeds that in the acute care hospital and that noise may be an important determinant of sleep quality [18]. However, all these studies were cross-sectional and did not investigate how much the inpatient setting impacts and changes sleep quality over the stay and whether such changes are related to patients’ characteristics. Such knowledge is, nevertheless, crucial since it would allow clinicians to better manage and optimize sleep quality as well as rehabilitation in such settings. In this preliminary, longitudinal study, our objective was to assess, for the first time, sleep quality at admission and discharge and to relate changes between time points to patients’ demographic and clinical characteristics.

## 2. Methods

### 2.1. Study Design

This prospective, longitudinal study was based on data obtained during assessments using validated scales (see Section 2.3) among patients recovering from critical illness or injury in an acute rehabilitation unit of CCHCH (Pomona, CA, USA). Assessments were performed at two time points just after admission and before discharge, approximately two weeks apart (13.5 ± 3.8 days). The demographic (age, gender, BMI, and ethnicity) and medical (reason for hospitalization, length of stay, pre-hospital living setting, and prior diagnosis of sleep disorders) variables of interest were extracted from patients’ medical records.

### 2.2. Study Population

Data were collected from February 2018 to August 2018. The inclusion criteria were the following: (1) able to complete verbal assessments; (2) 18 years old or more. The exclusion criteria were the following: (1) medical conditions that require intensive nursing care during the nighttime sleep period as determined by the attending physician (e.g., tracheotomy); (2) medical proxy unavailable to consent. All patients were admitted to the acute rehabilitation hospital from home, nursing facilities, or other hospitals. This prospective study was approved by the Institutional Review Board of the Casa Colina Hospital and Centers for Healthcare (CCHCH).

### 2.3. Outcome Measures

#### 2.3.1. Pittsburgh Sleep Quality Index

The Pittsburgh sleep quality index (PSQI) was the primary outcome measure of this study. It is a self-report questionnaire that assesses sleep quality over a time interval. The measure consists of 19 individual items, creating 7 components that produce 1 global score and takes 5–10 min to complete. The PSQI is intended to be a standardized sleep questionnaire for clinicians and researchers to use with ease and is used for multiple populations. Clinical studies have found the PSQI to be reliable and valid in the assessment of sleep disorders [19]. The higher the total score is, the worse the sleep quality is. A score of >5 has been related to poor sleep (with a sensitivity of 89.6% and a specificity of 86.5%) [20,21].

#### 2.3.2. PROMIS Scale v1.2-Global Health

The patient-reported outcomes measurement information system (PROMIS) provides clinicians and researchers access to reliable, valid, and flexible measures of health status that assess physical, mental, and social well-being from the patient perspective. This measure consists of 10 items to produce 1 global score and takes about 2–5 min to complete. PROMIS was established in 2004 with funding from the National Institutes of Health (NIH) as one of the initiatives of the NIH Roadmap for Medical Research [22].

#### 2.3.3. Beck Anxiety Inventory

The Beck Anxiety Inventory (BAI) is a 21 multiple-choice self-report inventory that is used for measuring the severity of anxiety symptoms. The questions included in this scale cover common symptoms of anxiety that occurred over the past week. The BAI is designed for individuals who are of 17 years of age or older and takes 5 to 10 min to complete. Each response is scored on a scale value of 0 (not at all) to 3 (severely). Higher total scores indicate more severe anxiety symptoms [23].

#### 2.3.4. Patient Health Questionnaire-9

The Patient Health Questionnaire-9 (PHQ-9), a tool specific to depression, scores each of the DSM-5-related criteria based on the mood module from the Primary Care Evaluation of Mental Disorders (PRIME-MD). It includes 9 questions measuring their depressive symptoms with response options ranging from 0 (not at all) to 3 (nearly every day). Higher scores suggest greater symptomology. The PHQ-9 takes about 5–10 min to complete and is both sensitive and specific in its diagnoses, which has led to its prominence in the primary care setting and in research studies [24].

#### 2.3.5. Sound Level and Sleep Disturbance Measurements

Nighttime sound levels (dB) and the average number of sleep disturbances within the acute inpatient rehabilitation setting were also obtained over four months. A total of 16 recordings were obtained once every week on the same day around 1 a.m. in the hall and inside the patient rooms using a sound pressure level meter.

### 2.4. Statistical Analysis

To assess the changes in sleep quality across time, repeated ANOVA measures were applied using PSQI total scores at admission and discharge as outcome variables. Interactions between changes in PSQI total scores and demographic/medical variables (age, gender, BMI, ethnicity, reason for hospitalization, length of stay, pre-hospital living setting, and prior diagnosis of sleep disorders) were tested. The association between patients’ mental health status and changes in the total PSQI scores was also assessed using the total scores of the PROMIS, BAI, and PHQ-9 at both admission (pre) and discharge (post). All results were considered significant at *p* < 0.05. Finally, descriptive statistics (average and standard deviation) were used to report nighttime sound levels and the average number of sleep disturbances within the acute inpatient rehabilitation setting.

## 3. Results

### 3.1. Participants

A total of patients participated in this preliminary study (age_mean_ = 61 ± 16 years old, 51% < 65 years old; BMI_mean_ = 30 ± 7 kg/m^2^; 51% female; 51% Caucasian). The average length of hospitalization was 18 ± 8 days. The reasons for hospitalization included orthopedic-related issue (28%; *n* = 10), spinal cord injury (28%; *n* = 10), stroke (20%; *n* = 7), and other (23%; *n* = 8). In this sample, 23% (*n* = 8) had prior sleep disorders (6 sleep apnea, 1 insomnia, and 1 narcolepsy) and 54% (*n* = 19) came from an acute care unit (see Table 1 for more details).

### 3.2. PSQI Changes Related to Demographics and Medical Variables

The main effect of the changes in the total PSQI scores at admission versus discharge was not observed [F (1, 34) = 0.317, *p* = 0.577]. However, a majority of patients (30 out of 35) obtained a score of > 5 at admission and only 20% of patients (*n* = 7) obtained a score of <5 at discharge. Using Chi-square as a secondary analysis, the change in proportion of patients with scores of >5 did not change significantly from admission to discharge (χ^2^ = 0.402; *p* = 0.526). Moreover, a significant interaction was observed between changes in total PSQI scores and diagnosed sleep disorders [F (1,33) = 12.861, *p* = 0.001, η^2^*_p_* = 0.280], where the presence of a disorder was associated with an improvement in sleep quality throughout the hospital stay. Post hoc analysis was performed to probe this significant interaction using the Bonferroni–Holm method and resulted in a significant difference between PSQI total scores at the two time points for patients with a diagnosed sleep disorder (p_holm_ = 0.009) but not for patients without a diagnosed sleep disorder (p_holm_ > 0.05).

Other medical variables such as pre-hospital living setting, length of stay, and reason for hospitalization were not related to changes in total PSQI scores (*p* > 0.05). Patients’ demographic variables such as age, gender, BMI, and ethnicity were also not related to changes in total PSQI scores (see Table 2 and Table 3 as well as Figure 1).

To assess differences in patients’ characteristics between groups (diagnosed sleep disorder versus no diagnosed sleep disorder) at the first time-point, a t-test was performed for continuous variables (i.e., age, length of stay, and BMI) and a Chi-square test for dichotomic variables (i.e., gender, ethnicity, pre-hospital living setting, and reason for hospitalization). We did not find any difference in terms of age (*t* = −0.541; *p* = 0.592), length of stay (*t* = −1.906; *p* = 0.065), BMI (*t* = −1.140; *p* = 0.263), gender (χ^2^ = 0.805; *p* = 0.369), ethnicity (χ^2^ = 6.328; *p* = 0.176), pre-hospital living setting (χ^2^ = 0.282; *p* = 0.595), or reason for hospitalization (χ^2^ = 0.952; *p* = 0.813) between groups (with versus without a diagnosed sleep disorder).

### 3.3. PSQI Changes Related to Mental Health

The PROMIS, BAI, and PHQ-9 total scores were not related at either time point (admission/discharge) to changes in PSQI total scores (see Table 4).

### 3.4. Sound Level and Sleep Disturbance

The mean nighttime sound collection level averages and peaks were 62.3 ± 5.1 dB and 86.1 ± 4.9 dB, respectively, and the average number of sleep disturbances was 2.6 ± 1.1.

## 4. Discussion

Our objective was to assess changes in sleep quality during an inpatient stay in a rehabilitation setting and to relate changes to patients’ demographic and clinical characteristics. According to our data, the majority of patients had poor sleep (PSQI > 5) at admission (86%) and at discharge (80%). None of the variables considered seemed to significantly impact the sleep quality of patients, except the presence of a sleep disorder. Indeed, based on our results, the sleep quality of patients with sleep disorders seemed to improve over the inpatient stay. These findings are novel, as previous studies have not assessed the sleep quality of patients in the presence of sleep disorders in such a clinical setting [25,26].

Sleep disorders (such as sleep apnea and also insomnia, narcolepsy, and restless leg syndrome) are more frequent in patients with spinal cord injury [27] and stroke [28] than in the general population and have been related to negative health outcomes such as cardiovascular risks and hypertension. Sleep deprivation and disorders have also been related to worsening of pain in patients with chronic pain and orthopedic issues [29]. The detection and management of these sleep disorders in an inpatient unit seems, therefore, crucial to ensure patients’ optimal recovery. As mentioned above, according to our results, the sleep quality of these patients improved over their stay. This result might appear as surprising. However, it might reflect specific care provided for diagnosed sleep disorders by respiratory therapists during the patients’ stay (e.g., CPAP in case of sleep apnea). Assessments or interventions were not provided to patients without a known sleep disorder. The poor sleep quality observed in these patients was, consequently, not addressed and stayed poor at discharge. It is worth it to mention that, even though their sleep quality improved significantly, most patients with sleep disorders (7 out of 8) still had poor sleep at discharge. One contributing factor was possibly the high nighttime sound level [15,30]. Excessive noise has previously been reported in intensive care units (ICUs) with documented peak noise levels more than those recommended by the Environmental Protection Agency for ICUs (45 dB during the day and 35 dB at night) with a mean noise level as high as 55–65 dB and peaks as high as 80 dB [31]. Morrison and coworkers (2003) found that higher than recommended sound levels were predictive of increases in heart rate, subjective stress, and annoyance in hospital nurses [32]. Finally, repeated disruptions due to patient care activities such as vital sign measurements, therapeutic interventions, or diagnostic procedures during daily resting time and also during the night might be even more disruptive to patient sleep [33]. The routine practice of collecting vital signs every four hours in hospitalized ward patients has been perpetuated since as early as 1893, but there is little evidence to support the necessity of this tradition [34]. Our findings have clinical implications and call for an increased assessment of sleep quality in a rehabilitation setting as well as highlight the importance of adapted management of sleep pathologies and of poor sleep when patients are admitted to such a setting. Moreover, our findings point to the need for future studies to investigate the implementation of quiet times for medical staff, patients, and family to help minimize noise levels and, hopefully, to improve sleep quality in these patients.

Our preliminary study has several limitations. First, this unicentric study had a relatively small sample size and calls for a multicentric study with a bigger sample to confirm our findings and increase generalizability. A significant difference in the main effect in sleep quality was not found but might have been due to the small number of time-points in this prospective design. Having more time points would allow for a greater detailed view of how sleep quality changes temporally. Future studies could also include measures, such as the functional independence measure (FIM), to relate changes in sleep quality to rehabilitation outcome [35]. Our preliminary study did not replicate previous results regarding demographic factors such as age, gender, BMI, and ethnicity. For instance, Madrid-Valero et al. 2017 found the prevalence of poor sleep quality is high among adults, especially women, and that there is a direct relationship between age and deterioration in the quality of sleep [36]. On the contrary to our study, this study was performed in a substantial sample (*n* = 2000) which led to a high effect resolution when compared to this study. In addition, BMI is an established predictor of poor sleep quality [37]; however, most of our patients were overweight (BMI = 25–29.9) or obese (BMI = 30 or greater), limiting the interpretation for this co-variate. Racial/ethnic differences in sleep quality are also documented [38], although our limited sample size prevented adequately-powered racial/ethnic comparisons. Thus, future research is needed to study associations between sleep disorders and sleep quality among communities of color within acute inpatient settings. Future studies could also add recording of light levels, as inappropriate exposure to light at night may cause melatonin secretion and adversely affect the biological clock and sleep quality [39]. Finally, our study did not show changes in quality of life [40], levels of anxiety [41], or depression [42], which could be explained by the short time window between assessments (2 weeks).

In summary, sleep is an essential component of health, and disrupted sleep is among the most common complaints of patients that survive a critical illness, impacting the rehabilitation process. Our preliminary study confirms that patients with a diagnosed sleep disorder should have their sleep quality closely monitored and also suggests that the sleep quality of inpatients admitted in an acute rehabilitation unit should be more systematically assessed and improved (if needed) throughout their stay. Our findings regarding nighttime sound levels and sleep disturbances advocate for the implementation of quiet time in such settings. However, further investigations are warranted. In the future, multicentric studies performed with a larger sample size will allow for better control of demographic, medical, and mental health related variables and should lead to a better understanding and management of the sleep quality in inpatient acute rehabilitation settings.

## Figures and Tables

**Figure 1 brainsci-13-00718-f001:**
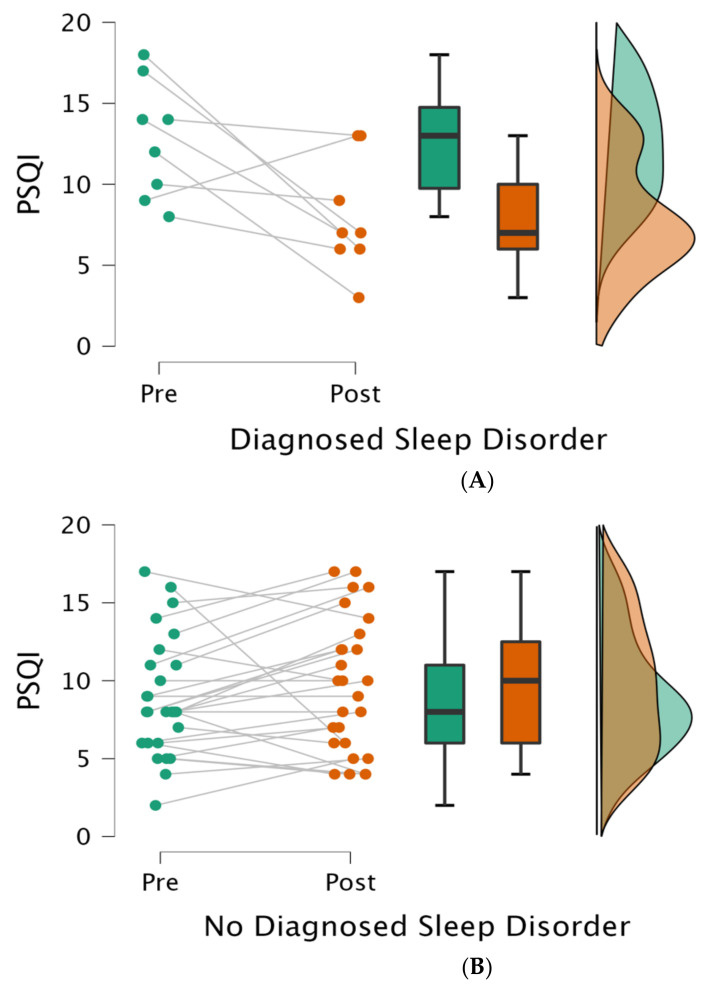
Sleep quality with or without a diagnosed sleep disorder at the two time points. Legend: Each panel (**A**,**B**) illustrates (from left to right) (1) individual data points, (2) averages (dark lines) with standard deviation (boxes) and 95% confidence intervals (brackets), and (3) variance at admission (pre; green) and discharge (post; red); PSQI = Pittsburgh sleep quality index.

**Table 1 brainsci-13-00718-t001:** Medical and demographic information.

ID	Age	Length of Stay	Gender	Ethnicity	Pre-Hospital Living Setting	Reason for Hospitalization	BMI	Diagnosed Sleep Disorder
1	55	13	Female	White	Acute Unit of Other Facility	Stroke	25	No
2	68	12	Female	White	Home	Other	25.91	No
3	78	10	Female	White	Home	Ortho	27.21	No
4	70	16	Male	White	Home	Ortho	26	No
5	67	13	Female	White	Home	Ortho	34.4	Yes
6	77	10	Female	Asian/Pacific Islander	Home	Other	20	Yes
7	25	12	Male	Hispanic or Latino	Home	Ortho	36.5	No
8	81	8	Female	Hispanic or Latino	Home	Ortho	27.6	No
9	44	26	Male	Asian/Pacific Islander	Acute Unit of Other Facility	Stroke	29.7	No
10	80	14	Female	White	Acute Unit of Other Facility	Other	26	No
11	79	13	Female	White	Acute Unit of Other Facility	Ortho	17.7	No
12	62	23	Female	Hispanic or Latino	Acute Unit of Other Facility	Stroke	30	No
13	61	27	Male	African American	Home	SCI	27.4	No
14	61	19	Female	White	Acute Unit of Other Facility	Stroke	39	No
15	92	15	Female	Asian / Pacific Islander	Acute Unit of Other Facility	SCI	27	No
16	67	15	Male	African American	Home	SCI	19.31	No
17	60	15	Male	White	Home	SCI	24.7	No
18	68	21	Female	Hispanic or Latino	Home	Ortho	27	No
19	80	14	Female	White	Acute Unit of Other Facility	Ortho	33.2	No
20	70	10	Male	Hispanic or Latino	Home	SCI	27.8	No
21	62	28	Female	White	Home	Other	na	No
22	65	11	Male	White	Acute Unit of Other Facility	Other	33.4	No
23	37	11	Female	White	Acute Unit of Other Facility	Other	26	No
24	55	32	Male	Hispanic or Latino	Acute Unit of Other Facility	SCI	39	Yes
25	63	16	Male	Asian/Pacific Islander	Acute Unit of Other Facility	Stroke	20.3	Yes
26	37	11	Female	Hispanic or Latino	Acute Unit of Other Facility	Other	47	No
27	57	19	Male	White	Acute Unit of Other Facility	SCI	33	Yes
28	23	35	Male	Hispanic or Latino	Acute Unit of Other Facility	SCI	22.8	No
29	50	23	Male	White	Home	SCI	32.3	No
30	65	23	Male	Other	Home	Stroke	26	Yes
31	59	28	Male	White	Acute Unit of Other Facility	SCI	40.9	Yes
32	66	38	Female	African American	Acute Unit of Other Facility	Ortho	46.7	Yes
33	50	20	Male	White	Acute Unit of Other Facility	Stroke	37.3	No
34	25	13	Male	Hispanic or Latino	Acute Unit of Other Facility	Ortho	33.76	No
35	70	16	Female	White	Home	Other	30.7	No

Legend: Age in years; length of stay in days; reason for hospitalization = ortho (orthopedic), SCI (spinal cord injury), BMI = body mass index (kg/m^2^).

**Table 2 brainsci-13-00718-t002:** Effect of demographic variables on sleep quality using repeated measure ANOVAs.

Effect	df	F	*p*	η^2^*_p_*
PSQI × Age	1, 33	0.936	0.340	0.028
PSQI × BMI	1, 32	0.755	0.391	0.023
PSQI × Gender	1, 33	0.076	0.785	0.002
PSQI × Ethnicity	1, 30	1.407	0.244	0.166

Legend: PSQI = Pittsburgh sleep quality index; age in years; BMI = body mass index (kg/m^2^).

**Table 3 brainsci-13-00718-t003:** Effect of medical variables on sleep quality using repeated measure ANOVAs.

Effect	df	F	*p*	η^2^*_p_*
PSQI × Diagnosed Sleep Disorder	1, 33	12.861	0.001 *	0.280
PSQI × Pre-Hospital Living Setting	1, 33	0.935	0.341	0.028
PSQI × Reason for Hospitalization	3, 31	0.146	0.931	0.014

Legend: PSQI = Pittsburgh sleep quality index; pre = admission; * significant results at *p* < 0.05.

**Table 4 brainsci-13-00718-t004:** Effect of mental health status on sleep quality using repeated measure ANOVAs.

Effect	df	F	*p*	η^2^*_p_*
PSQI × Pre PROMIS	1, 33	0.580	0.452	0.017
PSQI × Post PROMIS	1, 33	0.003	0.956	0.001
PSQI × Pre BAI	1, 33	1.961	0.171	0.056
PSQI × Post BAI	1, 33	0.095	0.760	0.003
PSQI × Pre PHQ-9	1, 33	1.431	0.240	0.042
PSQI × Post PHQ-9	1, 33	1.253	0.271	0.037

Legend: PSQI = Pittsburgh sleep quality index; PROMIS = patient-reported outcomes measurement information system; BAI = Beck Anxiety Inventory; PHQ-9 = patient health questionnaire-9; pre = admission; post = discharge.

## Data Availability

All data that support the findings of this study are available upon reasonable request to the corresponding author.

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
