# Peer review of "Sleep Quality Should Be Assessed in Inpatient Rehabilitation Settings: A Preliminary Study"

_brainsci, 2023, doi:10.3390/brainsci13050718_

Round 1
Reviewer 1 Report (Previous Reviewer 2)
thank you for addressing our concerns.
Author Response
and thank you for your constructive comments
Reviewer 2 Report (New Reviewer)
The current manuscript aims to assess changes in sleep quality during an inpatient stay in a rehabilitation setting through admission, discharge-admission sleep assessment . It concludes that Sleep quality should be assessed in inpatient rehabilitation settings. Please find my following comments:
1- The discussion did not explain the cause of the results or compare with current evidence
2- In inpatient settings, I think that there are several factors could affect sleep such as medications. Please discuss it
3- The discussion is written in the very long paragraphs. Please rewrite in shorter and informative form
Author Response
We thank the reviewer for his comments. We have highlighted in yellow the changes in the manuscript. As advised, the discussion was substantially modified to be shorter, to be structured in paragraphs that are more concise and to include a more recent literature. We also included the impact of pharmacological treatment as a limitation to our study. Finally, this version of the manuscript has been read by a native English speaker (BW). Orthographic and grammatical mistakes have been corrected.
Round 2
Reviewer 2 Report (New Reviewer)
The authors address all required comments
This manuscript is a resubmission of an earlier submission. The following is a list of the peer review reports and author responses from that submission.
Round 1
Reviewer 1 Report
Thank you for the opportunity to review your article.
This longitudinal study investigated the change in the severity of sleep problems at admission and discharge in inpatients, and found that the severity of sleep problems was improved in those without sleep disorders.
Although the longitudinal design is a little bit interesting, the study racks novelty and clinical implications. Furthermore, important confounds were not considered at analyses.This study should control for the disease that was the reason for hospitalization, since some physical diseases induce sleep problems, at least. The hospitalization period and room setting, light off time and so on also should be considered.
Reviewer 2 Report
dear colleagues, thank you for the submission.
some comments to consider:
the title is awkwardly formatted – consider making it more attractive and informative something like “an acute inpatient rehabilitation setting should assess sleep quality” please adjust and tailor
with small sample size this study needs to be pilot and un defines older persons as those aged 60 year or over. on many occasions it is defined as 65+. perhaps this need to be in title too as mean age is >61.
abstract – present data as % not n e.g. female 50% not 18
introduction missed some significant articles see e.g. https://pubmed.ncbi.nlm.nih.gov/25325580/ and https://pubmed.ncbi.nlm.nih.gov/23834036/ and https://pubmed.ncbi.nlm.nih.gov/32819888/
the introduction failed to justify this study, just because claiming it was not done does not justify the study.
“the primary goal of this study is to assess whether demographic and clinical factors are associated with sleep quality during the inpatient stay in a rehabilitation setting” please add specific objectives and hypotheses.
method is very difficult to follow with no details, many items not clear and some issues are difficult to correct at this stage. e.g., including young 23yrs and old 80 yrs people in the same analysis is problematic same thing about k/h of sleep disorders ideally these cases confound/bias results.
lack of sample size calculation and power analysis also makes results questionable.
no measure was used to identify severity of the disability.
table 1 is more of raw data than descriptive/summary results
analysis plan is questionable with two data points a paired sample t test would be easier to follow and reporting effect size cohen’s d will be essential
table 2 and table 3 report interactions and not main effects
figure 1 clearly showed above concern that decline in psqi is due to sleep disorder mainly not hospitalization
discussion needs to be improved after correcting the above issues
Reviewer 3 Report
I congratulate the authors for conducting very interesting research. My comments are in the methods chapter to introduce the research design first. I think the eauthors should add limitations and practical application of the study.